# Pre-hospital interventions in snakebite: A telephonic survey and follow up investigating snakebite envenoming from a tertiary care centre in Coastal Karnataka

Usha Wagle[1,2☉], Vrinda Lath[1,2,3☉], Vennila Jaganathan[4], Freston Marc Sirur[iD][1,2,3☉]*

1 Department of Emergency Medical Technology, Manipal College of Health Professions, Manipal Academy of Higher Education, Manipal, Karnataka, India, 2 Centre for Wilderness Medicine, Kasturba Medical College, Manipal, Manipal Academy of Higher Education, Manipal, Karnataka, India, 3 Department of Emergency Medicine, Kasturba Medical College, Manipal, Manipal Academy of Higher Education, Manipal, Karnataka, India, 4 Department of Biostatistics, Manipal College of Health Professions, Manipal Academy of Higher Education, Manipal, Karnataka, India

☉ These authors contributed equally to this work.
* Sirur.freston@gmail.com, Freston.sirur@manipal.edu

## Abstract

### Background

Snakebite envenoming remains a major public health issue, particularly in tropical and subtropical regions, where it disproportionately affects rural and socioeconomically disadvantaged communities. In India, especially in rural and tribal areas, inappropriate pre-hospital practices are common and often delay definitive medical care, contributing to poor outcomes and preventable complications.

### Objectives

This study aimed to assess community knowledge, attitudes, and practices related to snakebite first aid; describe the nature of medical and non-medical pre-hospital interventions; identify challenges in timely access to care; and examine the association between these factors and clinical outcomes.

### Methods

A prospective survey based cross sectional study of cases of snakebite envenoming enrolled in the VENOMS registry (CTRI/2019/10/021828) and hospital records was conducted. Patients who reported between August 2019 and July 2024 were identified from the VENOMS registry and Medical Records Department of Kasturba Medical College, MAHE, Manipal. After obtaining verbal consent, the victim, family member or first responder at the scene underwent a structured interview assessing pre-hospital interventions, awareness, attitude, present status of the victim, and

**Data availability statement:** All relevant data has been submitted as Supporting information.

**Funding:** The author(s) received no specific funding for this work.

**Competing interests:** The authors have declared that no competing interests exist.

concluded with a brief health education session. Geographic Information System (GIS) mapping was performed using QGIS (version-3.38.0).

## Results

Of the 273 patients, tourniquet was the most used intervention in civilian first response (70%). Patients who reached healthcare centre within 30 minutes were significantly associated with higher recovery rates and lower mortality (p = 0.003). Patients who used traditional healing methods had worse outcomes, higher mortality (7.3% vs. 0.9%) and disability (p = 0.023). Visiting three or more healthcare facilities was linked to increased disability, suggesting delays in definitive care (p = 0.003). Ambulances were utilized only in 46.1% of cases, where only 3 respondents reported using ambulance to reach a healthcare facility from home. In the remaining cases, ambulance was used only for inter-facility transfers. Furthermore, 22.71% of respondents reported unavailability of ambulance services in their area.

## Conclusion

Our study found that rapid transfer to definitive healthcare facility was associated with better outcomes, and that there are challenges in accessing emergency care for snakebite envenoming in our region. There is a pressing need to strengthen pre-hospital systems, improve public understanding of snakebite management, and enhance the capacity of Emergency Medical Services-trained and qualified paramedics through structured training programs like Bachelor and Master of Sciences in Emergency Medical Technology.

## Author summary

Snakebites remain a serious public health issue, especially in rural parts of India where access to timely medical care is limited. We followed 273 patients and found that most were first assisted by family or neighbors, not trained responders. Ambulance access was limited, and people mostly used private vehicles or auto-rickshaws to reach care. Many had to visit multiple hospitals before receiving definitive treatment, causing dangerous delays. A significant number also relied on traditional healing methods before seeking medical help. Patients who reached a hospital within the first hour had the best outcomes, while delays beyond 3–6 hours were associated with increased risks of disability and death. Our findings point to critical gaps in pre-hospital care, transport, and community awareness. Strengthening local health services, improving road and ambulance access, and empowering communities with the right knowledge and training could help reduce deaths and disabilities caused by snakebites.

## 1. Introduction

Snakebites have been classified as a neglected tropical disease by the World Health Organization (WHO) and are a major cause of mortality and morbidity in Lower-Middle Income Countries (LMIC), where access to timely medical care is often limited [1]. Globally, snakebite envenoming occurs in an estimated 1.8 to 2.7 million people annually, resulting in 81,000–138,000 deaths and approximately 400,000 cases of permanent disability [2–4]. Despite being a preventable cause of death and long-term disability, snakebite remains a neglected tropical disease, mainly affecting marginalized populations [5]. Epidemiological patterns of snakebite vary widely based on geographic, climatic, ecological, and cultural factors. For instance, the risk is notably higher during the rainy season when human-snake encounters increase due to agricultural practices, flooding, and changes in snake behaviour [6].

High snakebite morbidity and mortality have been linked to several factors, including the disproportionate stocking of antivenom, inadequate health services, delayed access to healthcare, inadequate training of medical personnel about managing snakebite, inadequate emergency infrastructure, the use of ineffective field/first aid measures, lengthy wait times before receiving appropriate treatment, and poor compliance with preventive measures [7]. The reliance on ineffective and potentially harmful traditional practices highlights a critical gap in public health education and access to emergency medical care. Factors such as limited healthcare infrastructure, scarcity of trained medical professionals in rural areas, and deeply rooted cultural beliefs contribute to delays in receiving the right treatment. These delays can lead to severe complications, prolonged hospital stays, increased financial burdens, and, in some cases, permanent disabilities [8].

Addressing the multifaceted challenges of snakebite management in India requires a comprehensive, multi-level approach that spans community education, healthcare system strengthening, and policy-level interventions. This study aims to evaluate the current landscape of pre-hospital interventions employed in snakebite cases in coastal Karnataka and analyse their impact on patient outcomes. By mapping out common pre-hospital practices, identifying factors contributing to delays in care, this research seeks to generate data-driven insights that can inform public health policies.

By focusing on strengthening the pre-hospital phase of snakebite management - the most crucial window for life-saving interventions - this study will contribute to reducing the burden of snakebite envenomation in India, ultimately improving both survival rates and long-term quality of life for affected individuals and their families.

## 2. Methods

### 2.1. Ethics statement

The study was revived and approved by Institutional Ethics Committee, Kasturba Medical College and Kasturba Hospital under approval number - IEC2:479/2024. The study involved human participants, including patients, family members and other first responders present at the scene. Each participant was contacted telephonically, and verbal informed consent was obtained in accordance with Informed Consent for Telephonic Interview format as approved by the ethics committee. In the case of child participants, verbal consent was obtained from the parent or legal guardian.

### 2.2. Study design

A prospective telephonic survey based cross sectional follow up was conducted of cases of snakebite envenoming at a single tertiary care centre in coastal Karnataka. All cases of snakebite envenoming were identified by screening cases enrolled in the VENOMS registry and Medical Records of KMC Manipal between 01st August 2019 and 31st July 2024.

The VENOMS registry is a prospective, Clinical Trials Registry – India (CTRI) registered (CTRI/2019/10/021828), multi-centric hospital-based registry on all envenomation. The study was cleared by the Kasturba Hospital Institutional Ethics Committee (IEC2:479/2024) (Annexure 1 in S1 Text) and registered in the CTRI (CTRI/2024/08/073073). A total of 578 cases were identified from Kasturba Hospital, of which 273 were included for analysis.

### 2.3. Participants

**Inclusion criteria.**

1. All patients who presented to the Emergency Department of Kasturba Hospital with a history of snakebite between 01 August 2019 and 31 July 2024, regardless of age, gender, or severity of envenomation, were included in the study.

**Exclusion criteria.**

a) Cases determined not to be snakebites based on clinical evidence, history, or syndromic assessment

b) Patients who did not provide consent for the telephonic interview

Patients enrolled from Kasturba Hospital, Manipal, Karnataka in the VENOMS registry and between 1st August 2019 and 31st July 2024 were screened, and those admitted with snakebite envenoming were included for telephonic interview. Contact numbers were obtained from medical records. Patients with envenomation by species other than snakes, and those who did not respond or consent to telephonic interview or were not present during the incident were excluded.

### 2.4. Development of survey tool

The questionnaire for the telephonic interview was developed following a literature review, expert opinion and gaps in Pre-hospital management of snakebite, as identified in the National Action Plan for Snakebite Envenoming (NAPSE) [9]. The questionnaire was reviewed and modified during focused groups discussions among the investigators. Key areas identified included civilian first response, access to healthcare, emergency medical services, primary healthcare, outcomes and One health. The following Table 1 summarizes the survey questions aligned with their corresponding thematic categories.

### 2.5. Methodology/Detailed description of data collection

After obtaining necessary permissions, the VENOMS registry and medical records were screened. Cases where culprit species were likely to be snakes were collected. Cases where culprit species could not be confirmed as snakes by evidence, history or syndrome were excluded. Telephone numbers were obtained from the medical records. Study participants included patients, family or other first responders present at the scene. Verbal consent was obtained from all participants. Each participant was contacted telephonically, and verbal consent was taken as per Informed Consent for telephonic Interview format. The Telephonic survey was conducted between September 2024 and January 2025. The respondent's presence at the scene was confirmed prior to interview. If they denied presence at the scene, they were asked to refer to another respondent who was present at the scene. Following this, consent was similarly obtained, and the questionnaire was administered. As a part of the questionnaire, the Participants were asked to report their Current disability status at the time of interview, which refers to functional limitations reported at the time of interview rather than disability immediately post discharge or over the study period. The outcome at discharge data were obtained from the VENOMS Registry and refer specifically to the patient's clinical status and outcome at the time of discharge from the hospitalization related to the snakebite event. Current knowledge, attitude and practice, which refers to the participants self-reported understanding and perception at the time of interview were also recorded. Since the interval between discharge and interview varied across participants, these responses represent retrospective perceptions rather than uniform post-discharge measures. The survey interview was limited to approximately 15 minutes, during which incident details were collected and survey questions were administered. The interview questions are included S1 Text and reflected in S1 Data. While the structure interview was completed within 15 minutes, any additional questions raised by participants regarding snakebite were addressed thoroughly, in their local language, and within a reasonable time at their convenience. For ethical purposes, health education on snakebite was also provided to participants (Annexure 3 in S1 Text). As recommended by the Institutional Ethics Committee, a social outreach component was incorporated into the process

**Table 1. Summary of survey questions with themes and gaps addressed.**

| Thematic area | Questions | Gaps addressed |
|---|---|---|
| **First Responder** | Who was the first responder following the snakebite incident? | Identifies who provides immediate care, addressing gaps in community and EMS preparedness. |
| **Pre-hospital practices** | Was any first aid treatment given before reaching the healthcare facility? If yes, then specify | Assesses first aid measures, a critical gap due to reliance on ineffective or harmful traditional practices. |
| **Access to healthcare** | Following the snakebite, how many healthcare facilities were visited before arriving at KH hospital? (Specify the order and level) | Captures healthcare-seeking patterns, inter-facility transfers and delay in reaching definitive care. |
| | After the bite, how long did it take to reach the first healthcare facility, Type of access and distance travelled to reach first centre? | Assesses delays in receiving care and barriers to geographic access. |
| | What was the mode of travel to receive the first response? (Specify the order and level) | Explores transport logistics and availability of single vs. multiple travel modes and ambulance utilization to reach healthcare facility |
| | What types of ambulance services are available in your area? | Assesses availability, types, gaps in utilization and understanding EMS. |
| | Name & location of the nearest healthcare facility? | Provides preliminary insights into spatial healthcare access and regional disparities, laying the groundwork for a separate, future study focused on geographic accessibility to medical care on the same topic. |
| | What were the challenges that were encountered in accessing medical care or reaching the hospital following the snakebite incident? | Identifies system-level and contextual barriers—including logistical, financial, and sociocultural—that hinder timely access to effective treatment. |
| **Pre-hospital care** | Was a healthcare worker present in the ambulance accompanying the patient? Yes/ No/ Not applicable If YES, then the type of worker and interventions in the ambulance. | Assesses quality of care during transit, addressing gaps in trained personnel presence and delivery of life-saving interventions during interfacility transfers. |
| | What medical interventions were done in the ambulance? | Captures the scope of emergency care provided en route, contributing to an understanding of EMS preparedness. |
| **One health/ Conservation** | After the snakebite incident, what happened to the snake that bit? | Explores human-animal interaction post bite, contributing to ecological surveillance |
| **Outcome** | Does the patient have any disability currently? | Assesses the long-term outcome and gaps in follow up care |
| **Community awareness** | I would like to ask certain steps on first aid can you confirm if it's correct or wrong? (Options provided in Annexure 2 in S1 Text) | Assesses baseline knowledge of evidence-based first-aid practices, addressing the need for targeted community education and behavioural interventions. |

both to uphold ethical standards by giving back to participants and to strengthen community awareness and public health response to snakebite. This component was independently validated by three external experts. This educational component was not used in any aspect of data analysis or interpretation. All clinical images (Figs 7–9) included under discussion section are original, captured by VENOMS Registry investigators, after obtaining informed consent from patients or their attendants. Fig 1 shows the patient recruitment process for the study.

## 2.6. Mitigating Interview bias

All participant interviews were conducted by the principal investigator, as a part of the postgraduate thesis project on which the study is based and was done with honesty, integrity and good ethical conduct. To minimize potential interview bias and recall bias during data collection, several strategies were employed. A structured questionnaire with predominantly closed ended questions was used to ensure uniformity of the response and reduce subjective interpretation. Wherever a reliable first responder or witness was unavailable at the time of the interview, efforts were made to identify and contact alternative credible sources of information, such as accompanying relatives. In instances where verbal accounts were uncertain, previously documented medical records, case sheets, or old reports were received wherever possible to validate and cross check the information obtained.

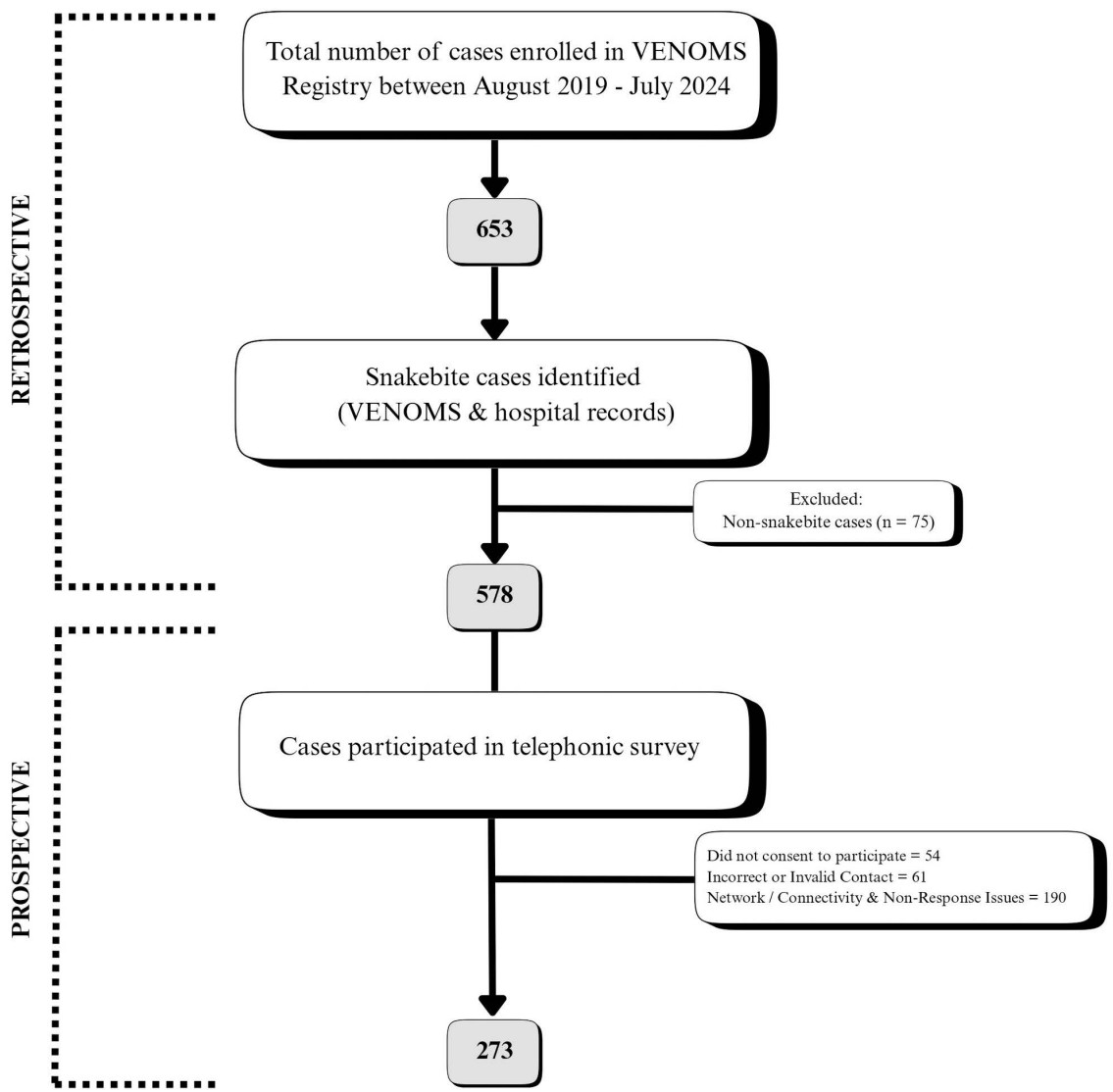

**Fig 1. Flow diagram of patient recruitment and inclusion in the study.**

## 2.7. Analytical tools

The responses were captured in Microsoft Excel and analysed using Microsoft Excel (Version 2505 Build 16.0.18827.20102) and Jamovi statistical software (version 2.4.11). Descriptive statistics were used to summarize the data. Continuous variables were summarized as mean ± Standard deviation (SD) and categorical variables as frequencies and percentages. Data were checked for normality using the Shapiro-Wilk test. For inferential analysis, Fisher's exact test was employed to examine associations between variables, as several cell counts were less than five. p-value of < 0.05 was considered statistically significant. Geographic Information System (GIS) mapping was performed using QGIS software (version-3.38.0).

## 3. Results

A total of 578 cases were identified retrospectively from the Registry and hospital records during the study period and subsequently contacted for the telephonic survey. Of these, 273 consented to participate in the survey. The survey was administered to eyewitnesses, reliable informers, or good samaritans who were directly involved in providing the first response and in transporting the patient to the hospital.

### 3.1. Patient demographics

Table 2 summarizes the patient demographics, circumstances of the bite, seasonal distribution, method of identification, species identified, and length of hospitalization.

### 3.2. First responder and response post bite

Table 3 illustrates the distribution of first responder and the type of response post bite. The term *first responder* refers to the civilian who provided the initial first aid. The first responders were most commonly family members (59%), followed by the patients themselves (29.3%), colleagues (8.8%) and public (2.9%). Among the first responses, the application of tourniquets (70%) was most frequently reported, followed by immediate transfer to a healthcare facility (21.6%) and the use of traditional healing practices (15%). Since most patients received more than one intervention, the cumulative percentage exceeds 100%, reflecting multiple interventions per individual.

**Association of pre-hospital interventions with outcome.** As summarized in Table 4, significant association was observed between the use of traditional healing and patient outcomes (p = 0.023). Patients who avoided traditional remedies had higher full recovery (70.7%) and lower mortality (0.9%) compared to those who used them (56.1% recovery, 7.3% mortality). Similarly, the number of healthcare facilities visited influenced outcomes (p = 0.003); patients visiting one or two centres had better recovery (75.9% and 72.2%), while visits to three or more facilities were linked to increased disability. Time to first healthcare contact also significantly impacted outcomes (p = 0.023); those reaching care within one hour showed the best recovery rates, while delays beyond 3 hours correlated with higher rates of disability, mortality, and Discharge against medical advice. Hemodynamic status on arrival was a strong predictor (p = 0.001); unstable patients had worse outcomes, with 15.4% mortality versus 0.4% among stable patients. Hemodynamic status was assessed using clinical indicators, specifically signs of shock and hypoxia. These parameters were chosen as they provide rapid, reliable and clinical information on the circulatory and respiratory status of the patients.

**Association of Tourniquet with Time to hospitalization.** Of the 191 patients who applied a tourniquet post bite, 57.6% arrived at a healthcare facility within 30 minutes, and an additional 27% within 1Hour as shown in Table 5.

### 3.3. Order and Level of healthcare facility visited

Table 6 highlights the healthcare facility utilization pathway that details patient transitions from initial contact to the final treating centre, Kasturba Medical College & Hospital (KMC) Manipal. A total of 87(31.85%) patients presented directly to our centre. The remaining patients initially presented to Primary, community or private hospitals/ clinics and were subsequently referred or self-referred to higher levels (tertiary centre) of care. One patient arrived after visiting four healthcare facilities, highlighting variability in referral patterns and potential delays in definitive care.

**Time taken to reach the first healthcare facility.** Fig 2 illustrates the time taken to reach the first healthcare centre. Of the 273 patients, 142 (52.0%) reached a medical facility within 30 minutes, 73 (26.7%) within 30 minutes to 1 hour, and 29 (10.6%) within 1–3 hours. Delays beyond 3 hours were observed in 29 patients of which 8 patients presented after 48 hours—one of whom sought care only after a month due to complications. Among the 273 patients who reached a healthcare facility, 107 (39.2%) received antivenom at the first point of care.

**Table 2. Demographics, Circumstances, Season, method of identification, Species identified and length of hospitalization.**

| Gender | | Method of identification | |
|---|---|---|---|
| Male | 182 (66.7%) | Photographic evidence | 43 (15.8%) |
| Female | 91 (33.3%) | Dead/live specimen brought to hospital | 33 (12.1%) |
| **Age** | | Photographic identification | 56 (20.5%) |
| 0-10 | 17 (6.2%) | Syndromic Identification | 141 (51.6%) |
| 11-20 | 20 (7.3%) | **Species identified** | |
| 21-30 | 31 (11.4%) | Russell's viper | 71 (26%) |
| 31-40 | 44 (16.1%) | Spectacled cobra | 20 (7.3%) |
| 41-50 | 47 (17.2%) | Saw scaled viper | 2 (0.7%) |
| 51-60 | 69 (25.3%) | Common Krait | 11 (4.0%) |
| 61-70 | 32 (11.7%) | Malabar pit viper | 6 (2.2%) |
| 71-80 | 10 (3.7%) | Hump nosed pit viper | 32 (11.7%) |
| 81-90 | 3 (1.1%) | Non-venomous | 61 (22.3%) |
| **Season of bite** | | Syndromic-hemotoxic | 56 (20.5%) |
| Summer | 65 (23.8%) | Syndromic- Neurotoxic | 6 (2.2%) |
| Monsoon | 104 (38.1%) | Unknown snake | 8 (2.9%) |
| Post monsoon | 50 (18.3%) | **Length of hospitalization** | |
| Winter | 54 (19.8%) | Min-Max (days) | 1 - 39 |
| **Circumstances of bite** | | | |
| Agricultural activity | | – | 89 (32.6%) |
| Walking | | On road during day | 17 (6.2%) |
| | | On road during night | 31 (11.4%) |
| | | In the fields | 9 (3.3%) |
| At residence | | Activity not specified | 22 (8.1%) |
| | | Outside residence | 26 (9.5%) |
| | | Sleeping inside residence | 3 (1.1%) |
| | | Household chores | 17 (6.2%) |
| Human Animal conflict situation | | Handling snake | 6 (2.2%) |
| | | Stepped on a snake | 2 (0.7%) |
| | | While chasing dog | 2 (0.7%) |
| | | Livestock handling | 1 (0.4%) |
| Playing | | Indoors | 15 (5.5%) |
| | | Outdoors | 2 (0.7%) |
| Construction/Industrial work | | – | 3 (1.1%) |
| Gardening | | – | 6 (2.2%) |
| Collecting firewood | | – | 8 (2.9%) |
| While in pond | | – | 2 (0.7%) |
| Urinating in open space | | – | 2 (0.7%) |
| Cleaning drains | | – | 1 (0.4%) |
| Vehicle related | | – | 2 (0.7%) |
| Activity not specified/ Not known | | – | 7 (2.6%) |

**Table 3. Frequency distribution of first responders following snakebite and types of initial responses provided to the patients.**

| Category | Type | Frequency |
|---|---|---|
| **FIRST RESPONDER POST BITE** | Family members | 160 (59%) |
| | Patient | 80 (29.3%) |
| | Colleagues | 24 (8.8%) |
| | General Public | 8 (2.9%) |
| **FIRST RESPONSE** | Tourniquet | 191 (70%) |
| | Immediate transfer to healthcare facility | 59 (21.6%) |
| | Traditional Healing methods | 41 (15%) |
| | Incision | 5 (1.8%) |
| | Bloodletting | 4 (1.5%) |
| | Scraping | 1 (0.4%) |

**Distance travelled to reach the first centre.** Of the 273 participants, 271 reported the distance travelled to reach the first healthcare centre, with an average of 13.17 km. Two participants were unsure of the distance. The minimum distance reported was 0.2 km, and the maximum was 80 km.

**Type of road used for transport.** Of the 273 patients, 267 (97.8%) reported accessing healthcare facilities via tar roads. A small number reported using motorable mud roads (5; 1.8%), and only one patient (0.4%) reported no accessible road.

**Mode of transport.** Table 7 presents the levels of transport used to reach the final treating centre. At the first level of transport, the most used mode was private cars (158 patients), followed by autorickshaws (63 patients). Ambulances were used by only 3 patients for initial transport; the remaining ambulance use was primarily for interfacility transfers (95 at second level, 61 at third level and 14th at fourth level).

**Presence of healthcare worker in ambulance.** Of the 273 patients who consented to participate, only 129 (47.3%) utilized ambulance services either to reach a healthcare facility or for inter-hospital transfers to higher-level centres. Among these, only 43 patients (15.8%) were accompanied by a healthcare provider during transport. Of these, 42 (15.4%) were accompanied by nurses and only 1 (0.4%) by a doctor, with possible misclassification of Emergency Medical Technicians (EMT) as nurses due to limited public awareness.

**Interventions in ambulance.** Of the 273 patients, 129 used ambulance services; among them, 6 (2.2%) received oxygen, 20 (7.3%) received Intravenous (IV) fluids, and 6 (2.2%) received both. The type of IV fluid, including possible ASV administration, could not be confirmed due to lack of documentation and absence of a pre-hospital data capture or telemedical exchange system.

### 3.4. Challenges faced in receiving medical care

Table 8 summarizes the challenges faced in receiving medical care. Human resource constraints (16.1%), infrastructure and logistics (6.7%) and community level barriers (10.6%) were the common challenges identified.

**Ambulance availability.** Table 9 highlights the ambulance availability in their area. Of the 273 patients, 62 (22.7%) reported that ambulances were unavailable in their area (as shown in Fig 3), while 22 (8.1%) stated they were unaware of ambulance availability

**Geospatial mapping of areas where ambulance services have been reported unavailable.** The geospatial mapping (Fig 3) highlights the reported locations across Karnataka where patients indicated that ambulance services were unavailable at the time of incident. These areas are predominantly concentrated in coastal and interior regions, particularly within Uttara Kannada, Udupi, and parts of Chikkamagaluru and Shivamogga district.

**Table 4. Associations between intervention, healthcare facility visited and stability with outcomes at the time of discharge/outcome of that hospitalization of the snakebite event (categorized as Death, Discharge with full recovery, Discharge with disability/Organ dysfunction and Discharge against medical advice).**

| Variables | Parameters | Outcomes at discharge | | | | Fisher's Exact Test p-value |
|---|---|---|---|---|---|---|
| | | Death (n=5) | Discharge with full recovery (n=187) | Discharge with disability/Organ dysfunction (n=30) | Discharge against medical advice (n=51) | |
| Traditional Healing methods | No | 2 (0.9%) | 164 (70.7%) | 23 (9.9%) | 43 (18.5%) | 0.023* |
| | Yes | 3 (7.3%) | 23 (56.1%) | 7 (17.1%) | 8 (19.5%) | |
| Number of healthcare facilities visited | 1 | 2 (2.3%) | 66 (75.9%) | 3 (3.4%) | 16 (18.4%) | 0.003* |
| | 2 | 2 (1.7%) | 83 (72.2%) | 8 (7.0%) | 22 (19.1%) | |
| | 3 | 1 (1.7%) | 31 (53.4%) | 16 (27.6%) | 10 (17.2%) | |
| | 4 | 0 (0.0%) | 6 (50.0%) | 3 (25.0%) | 3 (25.0%) | |
| | 5 | 0 (0.0%) | 1 (100%) | 0 (0.0%) | 0 (0.0%) | |
| Time to the first healthcare facility | 0 - 30min | 2 (1.4%) | 105 (73.9%) | 12 (8.5%) | 23 (16.2%) | 0.003* |
| | 30min - 1H | 0 (0.0%) | 51 (69.9%) | 7 (9.6%) | 15 (20.5%) | |
| | 1 - 3H | 0 (0.0%) | 19 (65.5%) | 5 (17.2%) | 5 (17.2%) | |
| | 3 - 6H | 0 (0.0%) | 3 (50.0%) | 1 (16.7%) | 2 (33.3%) | |
| | 6 - 12H | 0 (0.0%) | 4 (57.1%) | 0 (0.0%) | 3 (42.9%) | |
| | 12 - 24H | 1 (33.3%) | 0 (0.0%) | 1 (33.3%) | 1 (33.3%) | |
| | 24 - 48H | 1 (20.0%) | 1 (20.0%) | 2 (40.0%) | 1 (20.0%) | |
| | >48H | 1 (12.5%) | 4 (50.0%) | 2 (25.0%) | 1 (12.5%) | |
| Stability on ED Arrival | Stable | 1 (0.4%) | 173 (70.0%) | 28 (11.3%) | 45 (18.2%) | 0.001* |
| | Unstable | 4 (15.4%) | 14 (53.8%) | 2 (7.7%) | 6 (23.1%) | |

*Indicates statistically significant difference (p<0.05). Fisher's exact test was used for associations.

**Table 5. Association of Tourniquet with Time to hospitalization.**

| Variables | Parameters | Time to hospitalization | | | | | | | | Fisher's Exact Test p-value |
|---|---|---|---|---|---|---|---|---|---|---|
| | | 0-30min | 30min -1H | 1-3H | 3-6H | 6-12H | 12-24H | 24-48H | >48H | |
| Tourni-quet | No | 32 (39%) | 21 (25.6%) | 8 (9.8%) | 4 (4.9%) | 4 (4.9%) | 3 (3.7%) | 4 (4.9%) | 6 (7.3%) | <0.001* |
| | Yes | 110 (57.6%) | 52 (27.2%) | 21 (11%) | 2 (1.0%) | 3 (1.6%) | 0 (0%) | 1 (0.5%) | 2 (1.0%) | |

*Indicates statistically significant difference (p<0.05).

**Table 6. Frequency distribution of level & order of healthcare facilities visited.**

| Level of healthcare facility | Order of healthcare facility visited | | | | |
|---|---|---|---|---|---|
| | 1st Order (N = 273) | 2nd Order (N = 186) | 3rd Order (N = 71) | 4th Order (N = 13) | 5th Order (N = 1) |
| Primary Health Centre (PHC) | 57 | 4 | 1 | 0 | 0 |
| Community Health Centre (CHC) | 1 | 0 | 1 | 0 | 0 |
| Taluka hospital (Town) | 36 | 13 | 1 | 0 | 0 |
| District hospital | 21 | 34 | 5 | 1 | 0 |
| Private clinic | 23 | 0 | 0 | 0 | 0 |
| Private hospital | 48 | 20 | 5 | 0 | 0 |
| KMC Manipal (tertiary) | 87 | 115 | 58 | 12 | 1 |

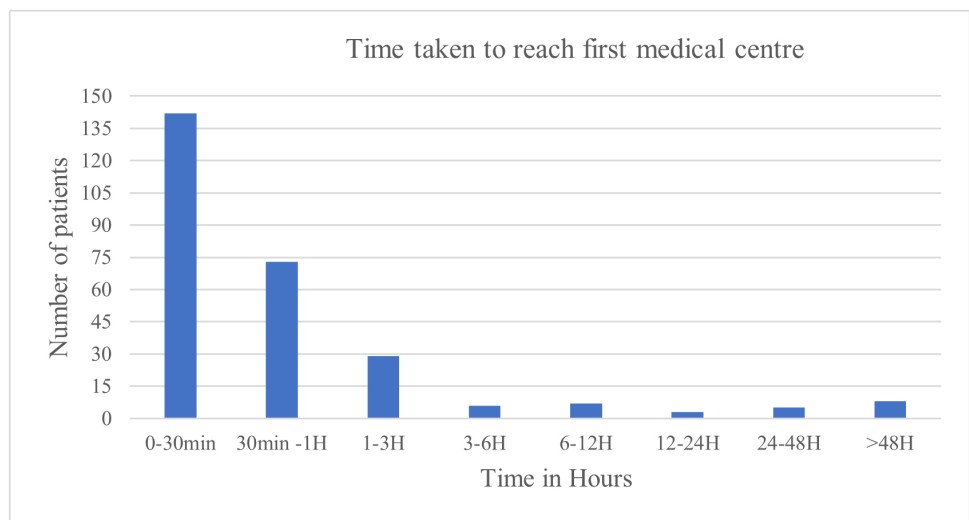

**Fig 2. Bar chart showing time taken to reach first Healthcare Centre.**

**Table 7. Frequency distribution of level of transport used.**

| Mode of Transport | 1st level (N = 273) | 2nd level (N = 191) | 3rd level (N = 77) | 4th level (N = 18) | 5th level (N = 1) |
|---|---|---|---|---|---|
| Bike | 43 | 1 | 0 | 0 | 0 |
| Car | 158 | 70 | 14 | 3 | 1 |
| Autorickshaw | 63 | 14 | 2 | 1 | 0 |
| Ambulance | 3 | 95 | 61 | 14 | 0 |
| Bus | 3 | 10 | 0 | 0 | 0 |
| Walk | 2 | 0 | 0 | 0 | 0 |
| Truck | 1 | 1 | 0 | 0 | 0 |

**Table 8. Frequency table for challenges faced in receiving medical care.**

| Category | Specific challenges | Frequency (%) |
|---|---|---|
| No challenges Reported | No challenges faced | 164 (60.1%) |
| Human Resource Constraints | • No Snakebite treating expert in local facility<br>• No healthcare worker at Primary Health Centre | 44 (16.1%) |
| Infrastructure and logistics | • Lack of ambulance services<br>• No motorable road<br>• Poor communication network | 18 (6.7%) |
| Healthcare facility limitations | • Non availability of ICU beds<br>• Non availability of ASV<br>• Advised Haemodialysis | 7 (6.2%) |
| Community-level barriers | • Lack of awareness | 29 (10.6%) |
| Healthcare system negligence | • Negligence at medical facility | 1 (0.4%) |

**Table 9. Frequency distribution of ambulance availability.**

| Ambulance Availability | Frequency (%) |
|---|---|
| Private ambulance – Basic Life Support | 15 (5.5%) |
| Private ambulance – Advanced Life Support | 34 (12.5%) |
| Government ambulance – Basic Life Support | 17 (6.2%) |
| Government ambulance – Advanced Life Support | 41 (15.0%) |
| Both Government & Private ambulance | 82 (30.0%) |
| Not available | 62 (22.7%) |
| Does not know | 22 (8.1%) |

### 3.5. Current disability during telephonic interview

Fig 4 depicts the current disability during call. Of the 273, a total of 11 (4%) deaths were reported, of which 2 occurred in the hospital. Three deaths were attributed to causes unrelated to the snakebite, while the remaining were reported as result from snakebite-related complications.

### 3.6. Actions taken regarding the snake post bite

Fig 5 illustrates post bite handling of snakes. In 159 (58.2%) cases, the snake left the scene without being captured. 8 patients (2.9%) brought the live snake to the hospital, while 50 (18.3%) reported killing the snake—of these, 31(11.4%) discarded the specimen and 19 (7%) brought the dead snake to the hospital for identification. In 56 cases (20.5%), the snake was not seen at all.

### 3.7. Current Knowledge and attitude

Current attitude and knowledge of the patients (Fig 6) were assessed by asking a series of questions that included both appropriate and inappropriate response. A majority of 217 participants (79.5%) reported applying a tourniquet, while nearly all—269 participants (98.5%)—acknowledged the importance of reassurance and anti-snake venom administration. Correct knowledge of limb immobilization was noted in 79 (28.9%) participants, while 163 (59.7%) were unsure. Traditional or Ayurvedic remedies were considered by 73 (26.7%), and 47 (17.2%) supported incision or bloodletting, though 221 (81%) participants rejected these practices. Capturing evidence of the snake (e.g., photograph) was recognized as important by 269 (98.5%), while 113 (41.1%) stated they would kill the snake post-bite. Only 17 (6.2%) contacted community health workers (e.g., Accredited Social Health Activist workers).

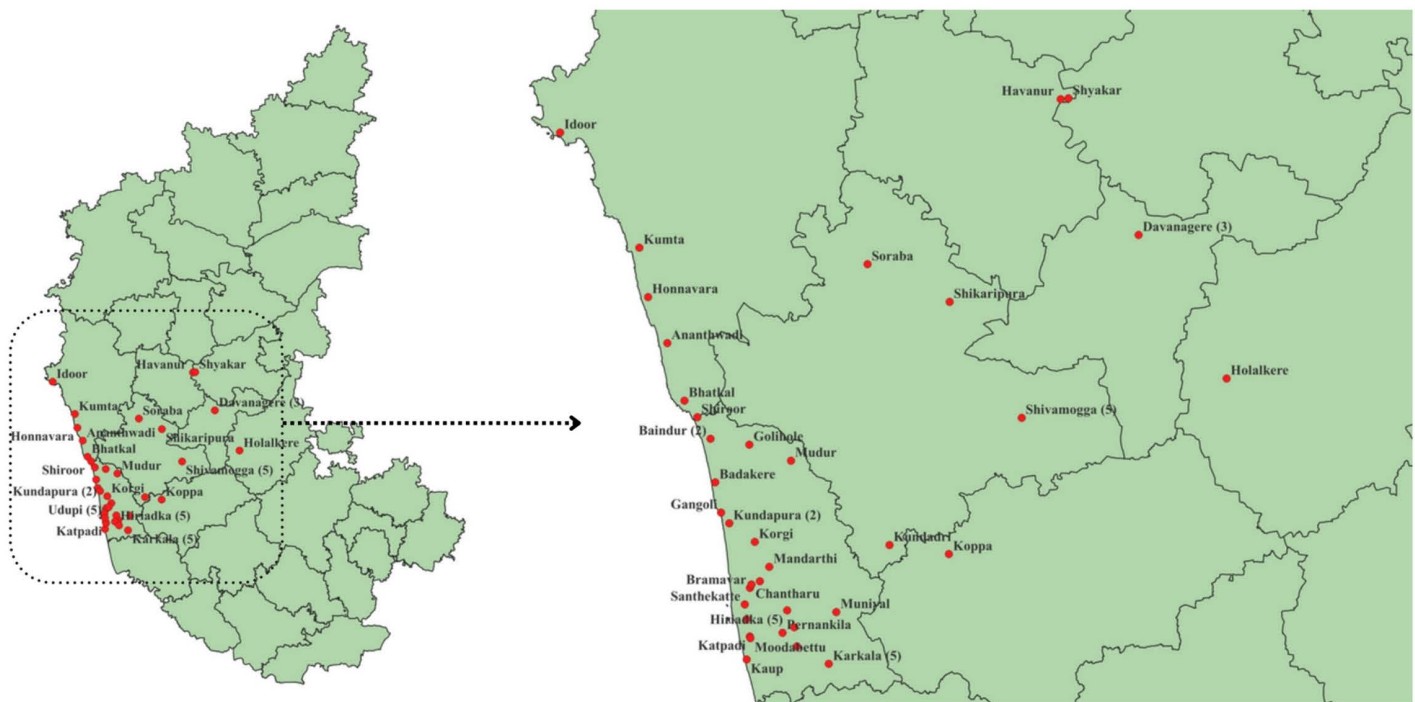

**Fig 3. Mapping of areas where ambulance services were reported as unavailable.** The base shapefile used for district boundaries was sourced from the GitHub repository (District.zip, available at: https://github.com/JKAY3366/karnataka_pop_by_hr), which is licensed under CC BY 4.0. All maps were created solely using shapefile from GitHub. All mapping and spatial analyses were performed using QGIS software (https://qgis.org/). No additional base layer was incorporated from QGIS.

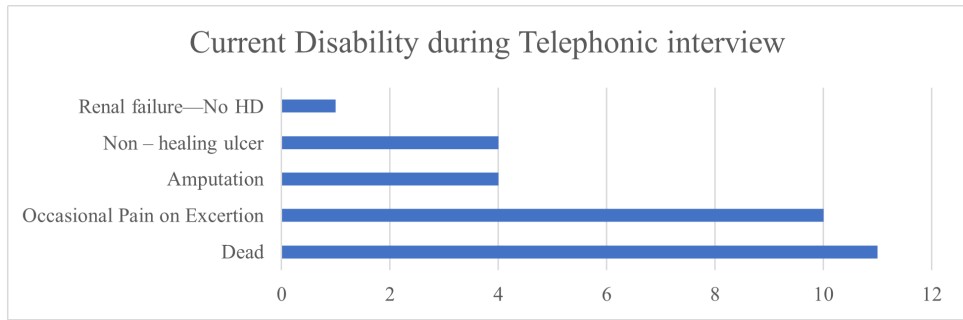

**Fig 4. 2-D bar chat representing Current Disability during Telephonic interview.**

## 4. Discussion

### 4.1. Civilian first response

As illustrated in Table 3, the findings highlight the critical role that civilian first responders, especially family members (59%) play in managing snakebite incidents, particularly in rural and remote areas where pre-hospital services are limited. None of the first responses in our study were Accredited Social Health Activist (ASHA) workers or trained healthcare providers; instead, the vast majority were civilians, often the close relative of the victim. In 29.3% of cases,

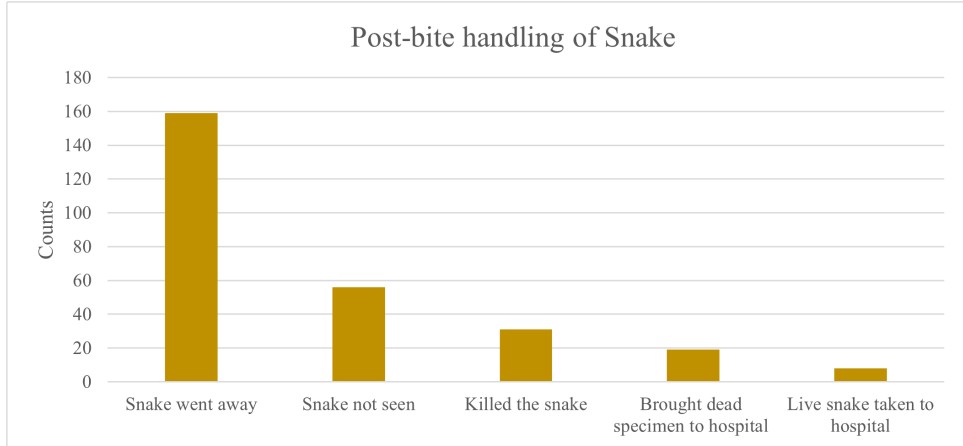

**Fig 5. Bar chart representing the frequency of post-bite snake handling.**

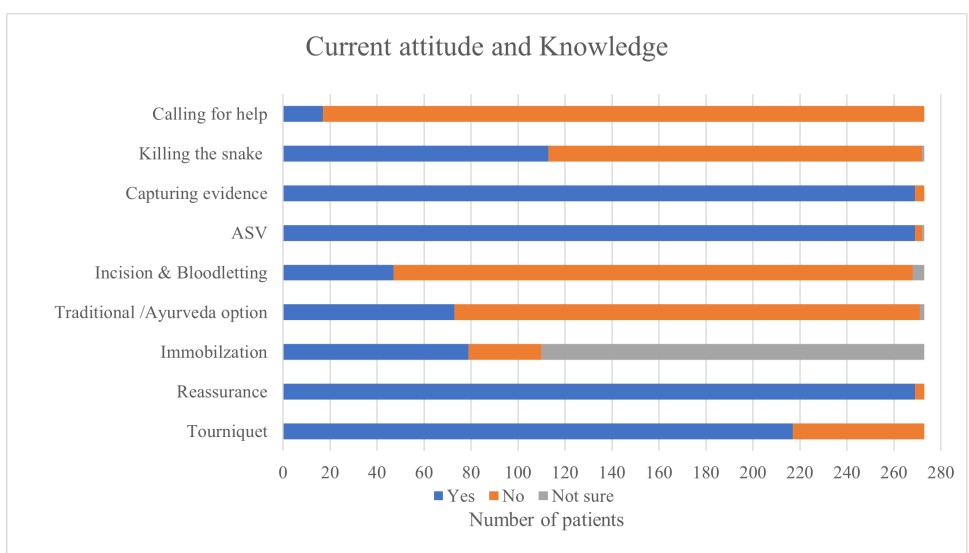

**Fig 6. 2-D car chart representing current attitude and knowledge towards snakebite.**

patients had to extricate themselves from the scene of the incident, without any immediate assistance. In these cases, it is a major challenge particularly if the patient becomes critically ill and cannot extricate themselves, which can result in prolonged delays in receiving care also known as time to treatment latency. The absence of trained first responders at the scene not only limits the effectiveness of the initial response but also increases the likelihood of wrong practices, such as incisions or traditional healing practices. This further underscores the urgent need to train and upskill civilian first responders in snakebite-prone areas, equipping them with essential knowledge and skills for timely and appropriate care. Evidence from community-based training and education has shown positive results. A retrospective study from Nepal demonstrated that first aid training in snakebite management led to a reduction in overall mortality [10]. Similarly, a recent study highlighted that multifaceted community health education programs can substantially reduce snakebite related deaths, disabilities and the associated socioeconomic burden. Together, these findings underscore

that structured educational interventions at grassroot level can play a pivotal role in strengthening the chain of snakebite survival [11]. Therefore, integrated community-level training and awareness, fostering intersectoral collaboration are essential steps in strengthening the emergency response system, particularly in snakebite-prone areas and regions with limited healthcare access.

### 4.2. Pre-hospital practices

The findings of this study highlight critical gaps and challenges in the pre-hospital management of snakebite cases in India. Despite increasing awareness of snakebite envenomation as a public health emergency, the reliance on inappropriate first aid practices such as the use of tourniquet, traditional healing measures and harmful interventions like incisions and bloodletting persists (as shown in Table 3). These practices are not only outdated but also delay definitive care, often leading to avoidable complications and poorer outcomes.

**Tourniquets.** As mentioned in the results, tourniquet application was the most reported first response, with one instance occurring within a healthcare facility. Although historically recommended as a part of fit aid care for snakebites, [12] evolving clinical evidence and current guidelines no longer support their use due to the potential for serious complications such as restricted blood flow, compartment syndrome, ischemia and gangrene [13]. Despite this, the continued use of tourniquets suggests need not only for training but also for reteaching - to actively unlearn outdated and potentially harmful practices that are still commonly deployed in communities. Given the retrospective nature of our data and its reliance on telephonic reporting, no conclusions can be drawn regarding the clinical impact of tourniquet use. A significant limitation is the lack of granular details on application parameters such as tightness and duration of the intervention. Effective (tight) tourniquets can prevent the venom absorption but can cause severe limb damage. A study reported a case where prolonged tourniquet application led to ischemic necrosis, necessitating limb amputation [14]. Non-effective (loose) tourniquets do not restrict venom movement, making them medically ineffective but less harmful, and therefore, clear inferences could not be drawn here regarding the use of tourniquet. Because our dataset cannot distinguish between these variations, inferential interpretation was not carried out.

This limitation is consistent with findings from prior systematic reviews, which have not demonstrated any clinical benefit of tourniquets as a first aid measure of snakebite and instead report higher risks of local envenoming [15].

Our finding that more than 50% of tourniquet users (Table 5) reached healthcare within 30 minutes suggests that early hospitalization, rather than tourniquet use, likely influenced improved outcomes. Additionally, this study includes all forms of envenomation – hemotoxic (vipers) and Neurotoxic (elapids) which differ in pathophysiology and responsiveness to the first aid may have shaped the outcomes. Tourniquet use in elapid bites has been reported to delay the onset of neurotoxicity [16]. Thus, outcomes observed are likely influenced by multiple confounding factors, rather than attributable solely to tourniquet application. A prospective study would be necessary to evaluate these factors more robust and generate reliable inferences. Representative images of patients with pre-admission tourniquet application are shown in Fig 7. Although the continued use of tourniquets is widely discouraged in modern snakebite management guidelines, its persistence in the community remains a concern. While we do not have confirmed evidence regarding erroneous first-aid teaching in schools or universities within Karnataka, the possibility of such outdated practices being perpetuated through informal learning channels cannot be excluded. Similar issues have been documented in Nepal, where harmful first-aid interventions continue to be taught in formal academic settings [17]. In our context, the persistence of tourniquet use may be influenced by long-standing community practices, reliance on "natural intelligence" passed across generations, and variable exposure to accurate first-aid information. Therefore, improved access to validated educational materials and periodic principles could play a crucial role in discouraging harmful practices such as tourniquet application in snakebite cases.

**Traditional healing measures.** This response was used in 41(15%) cases. These measures included application of black stone with a belief that it absorbs the venom; consumption of neem leaves (*Azadirachta indica*), black pepper (*Piper nigrum*), salt, alcohol, herbal decoctions; topical application of herbal ointments; turmeric etc., as shown in the

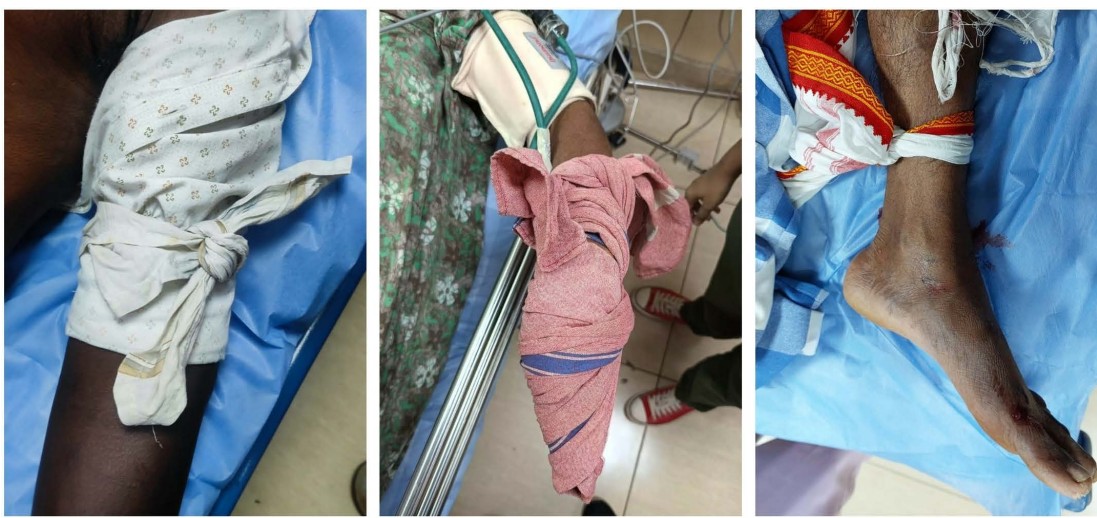

**Fig 7. Images of the bitten limb with tourniquet.**

Fig 8. Traditional healing methods were significantly associated with increased mortality and disability (p = 0.023) as shown in Table 4. Our findings are consistent with previous studies that have linked the use of traditional medicine to poorer outcomes in snakebite cases [18,19]. While traditional medicine may have no role in emergency care, it may hold potential value in managing chronic complications such as ulceration or tissue necrosis, when practiced with appropriate quality and safety. Certain plant-based remedies used in traditional healing may contribute to drug discovery efforts in the long term. Despite current guidelines prohibiting the use of shamans and traditional healers in snakebite care, their use persists. This persistence may be attributed to factors such as accessibility, cost-effectiveness, cultural beliefs, and trust in indigenous practices. Addressing this issue requires not only regulatory enforcement but also community education to shift harmful practices while respecting cultural contexts; additionally, strategically engaging traditional healers by educating

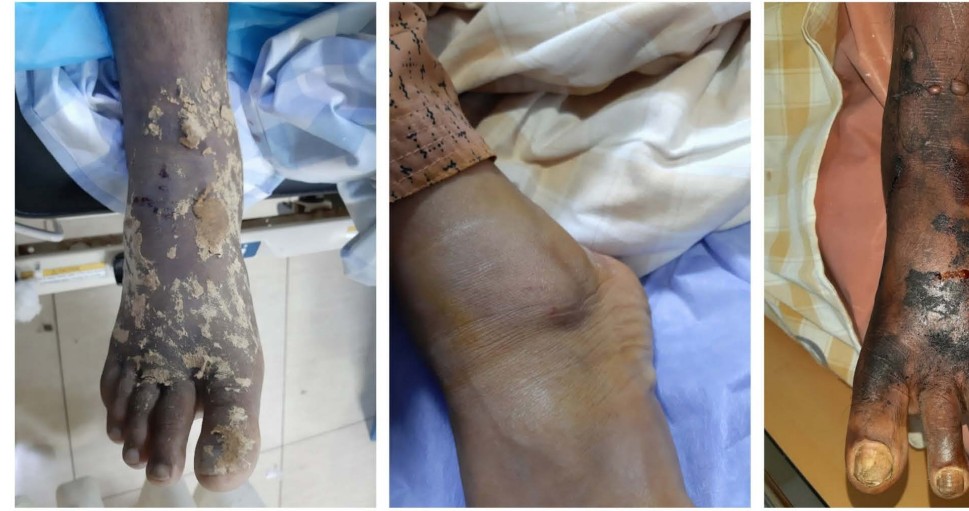

**Fig 8. Images of bitten limbs with topical application of herbal remedies.**

them to promptly refer suspected snakebite cases to formal healthcare facilities could serve as a pragmatic harm-reduction approach that bridges the gap between community trust in traditional systems and the need for timely care.

**Incision and bloodletting.** Practices such as incision and bloodletting were previously included in older first-aid teachings for snakebites, rooted in the belief that removing blood could expel venom [20–22]. These outdated methods have persisted across generations and continue to be used in some communities, highlighting a critical need for deletion of outdated knowledge and relearning of evidence-based practices. Despite their historical use, these methods lack scientific validation and are now strongly discouraged in current guidelines. They carry significant risks, including severe bleeding, secondary infection, and increased likelihood of sepsis, all of which can contribute to prolonged recovery and worsened outcomes. Fig 9 depicts cases where patients presented with incisions made prior to hospital admission

Other methods such as suction devices, electric shock therapy, and commercially marketed snake repellents are also prevalent in many communities. Suction devices, once recommended in early guidelines, have since been shown to offer no clinical benefit in venom removal and may even damage surrounding tissues [21,22]. The relevance of these to health seeking behaviour was not directly evaluated in our study, and none of the participants reported using repellents. However, we have included a discussion on repellents to highlight the issue of misinformation and misleading claims. Various snake repellent products ranging from sprays to electronic emitters—are openly and legally marketed despite lacking robust scientific evidence of efficacy. Studies have shown that snakes remained unaffected by the repellent [23]. The continued availability and promotion of such products not only mislead the public but also delay appropriate medical care, contributing to poorer outcomes.

As shown in Fig 6, the current attitudes and knowledge of participants in managing snakebite incidents highlights significant gaps in awareness regarding appropriate first-aid measures and the importance of early and timely transfer to a healthcare facility. These findings align with the interventions reported, as 70% of patients had applied a tourniquet post-bite, and 79.5% of participants indicated they would still prefer using a tourniquet in the event of a snakebite. One potential point of intervention is the healthcare facility itself, where patients and caregivers can be provided with appropriate education on evidence-based snakebite management. However, the continued reliance on harmful practices even after

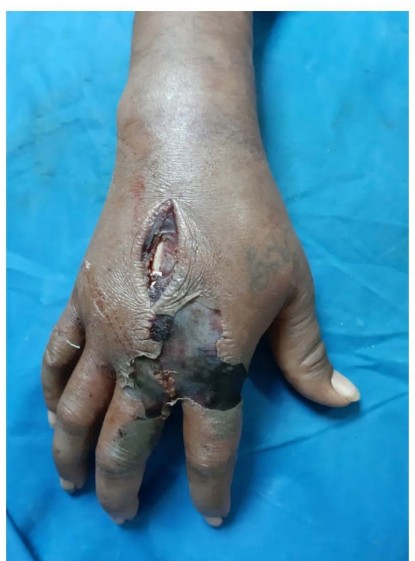 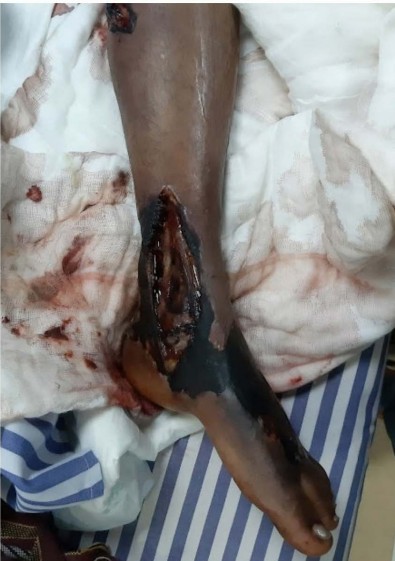

**Fig 9. Images of the bitten limbs with preadmission incisions.**

hospital visits suggests either a missed opportunity for such education or poor acceptance and retention of the information provided—highlighting the need for more effective and culturally sensitive communication strategies.

### 4.3. Access to emergency healthcare

The study revealed a concerning trend of multiple facility visits by patients prior to receiving definitive care for snakebite at a tertiary centre as shown in Table 6. The number of healthcare facilities visited had a significant impact on outcomes (p = 0.003) as presented in Table 4, with patients who visited three or more centres experiencing a higher disability rate (27.6%) and lower percentage of full recovery (53.4%). Of the 273 patients, 68.13% (186) visited more than 1 centre, 26% (71) visited more than 2 centres, 4.76% [13] visited more than 3 centres, 0.37% [1] visited more than 4 centres. The referral pathway here involved multiple transitions—starting at the district hospital, followed by two successive private hospitals, with a subsequent visit to the district hospital before reaching our treating centre. Studies have shown that early referral to higher facility helps in reducing mortality rates [24]. Similarly, patients time to first healthcare centre had a significant impact on outcomes (p = 0.003) as presented in Table 4; those reaching care within one hour showed the best recovery rates, while delays beyond 3 hours correlated with higher rates of disability, mortality. A visual representation of the time to first healthcare facility is provided in Fig 2. However, reaching the "First healthcare facility" does not necessarily equate to receiving definitive care. Although 107 patients received antivenom at their first healthcare facility, many others encountered barriers that prevented timely administration of ASV. 44(16.1%) had reported human resource constraints such as the absence of snakebite treating experts or lack of available healthcare workers at local primary healthcare centres Similar patterns were observed in a study from rural north-eastern Nigeria, where prolonged delays in accessing care, particularly among patients living ≥100 km from the treating facility were associated with severe envenomation and higher rates of poor outcomes, including death [25]. This underscores that it is not merely the time to hospitalization, but rather the time to definitive care that significantly influences outcomes in snakebite cases. Our study suggests that under-equipped healthcare facilities may inadvertently contribute to worse outcomes, emphasizing the urgent need for targeted infrastructure upgrades, staff training, and effective triaging protocols. Strengthening government hospitals to provide appropriate initial management at the primary level while ensuring timely referral to higher centres when needed is essential. Furthermore, this highlights the critical role of emergency medical services (EMS) and ambulance systems in bridging the gap, by facilitating proper triage and transport to the most appropriate facility.

Many peripheral or remote healthcare facilities in the region typically function with limited antivenom stock and constrained workforce capacity, as also highlighted in national program documents such as the National Action Plan for Prevention and Control of Snakebite [26]. In such settings, patients may receive only an initial or limited stock of antivenom before being referred to higher centres for continued management, and a result, delays in receiving definitive treatment may occur even when the first healthcare facility is reached early, and this remains an important contextual factor influencing patient outcomes.

Furthermore, research indicates that only 22% of snakebite victims seek treatment at health facilities, and a mere 7% of snakebite deaths are officially recorded, highlighting significant gaps in the healthcare system's ability to manage snakebite cases effectively [27].

In time-sensitive emergencies such as trauma and stroke, there are well-established guidelines for timely referral to higher-level or specialized centre—for example, Level I trauma centre or facilities equipped for thrombolysis [28,29]. Similarly, the development of structured referral pathways for snakebite cases is crucial to improving patient outcomes. Strengthening the capacity of primary healthcare centres and ensuring timely referral to higher-level facilities can reduce the need for patients to visit multiple centres, thereby improving overall outcomes.

Despite free treatment being available at government facilities, many patients preferred private referral centres, reflecting a lack of public confidence in the public healthcare system—likely due to perceived gaps in manpower, infrastructure, or medicine availability. This also highlights gaps in formal education - first aid and snakebite awareness are systemically

integrated into school or university curriculum. Strengthening formal education and public awareness on snakebite management, including orientation on appropriate referral facilities where antivenom is available, may improve trust in public healthcare and guide patients to the right points of care. Programs such as Bachelors and Master's in Emergency Medical Technology in this region place strong emphasis on snakebite as it is a neglected tropical disease with significant impact on the Indian population, particularly in remote areas. These programs were not previously available in this region, and Manipal Academy of Higher Education is the first institution in Coastal Karnataka to establish such a program in the year 2019. The National Programme for Prevention and Control of Snakebite Envenoming was launched to address these gaps by improving surveillance, establishing regional venom centres, and enhancing healthcare professionals' capacity. Early stabilization at Primary Health Centres followed by timely referral has been shown to significantly improve survival and reduce long-term disability [26].

### 4.4. Challenges in emergency transport: Accessibility and infrastructure gaps

In remote and underserved regions, multiple factors impact timely snakebite care. Challenges include limited awareness, physical accessibility, road conditions, and terrain that affect both direct patient transport and the functioning of ambulance services. Additionally, inconsistent telephone network coverage hampers emergency coordination, especially in areas with poor connectivity and infrastructure. These access-related barriers significantly delay care and highlight the need for targeted infrastructural improvements. Human resource constraints (16.1%), infrastructure and logistics (6.7%) and community level barriers (10.6%) were the common challenges reported as shown in Table 8.

While ambulance usage was more prominent during secondary and tertiary interfacility transfers, private vehicles - such as cars and autorickshaws - remained commonly used modes of transport across all levels of care, as shown in Table 7. This trend underscores a complex interplay of factors, including patient and caregiver preferences, perceived or real limitations in ambulance availability. Autorickshaws were used by 63 patients (23.1%) at the first level (as shown in, mainly because they are easily available, affordable, and can navigate both rural and remote areas quickly. Additional factors in transport choices is the patient's behaviour, as many would prefer taking a mode of transport which is easily accessible rather than waiting for an ambulance due to the urgency driven by fear and anxiety. A study highlighted that autorickshaws, were the predominant mode of prehospital transport for road traffic injury victims, accounting for 53.2% of cases [30]. Given this, capacity building of these autorickshaws and facilitating their development as first responders for snakebite may be a solution for a Low- and Middle-Income Country like India.

The absence of standardized documentation and telemedical communication during ambulance transport hampers continuity of care. Without clear records of in-transit interventions, such as antivenom or intravenous fluids, receiving facilities face treatment delays, underscoring the need for integrated pre-hospital data systems.

As shown in Table 9, 62 (22.7%) patients reported unavailability of ambulances in their area, and the geographic distribution of this finding is illustrated in Fig 3. However, this perception warrants further investigation to determine whether ambulances are truly inaccessible or if the response reflects a lack of awareness about available emergency services or how to access them.

Road infrastructure in rural and remote areas, often marked by narrow, unpaved, or poorly maintained roads, significantly delays ambulance response times, making autorickshaws a more feasible mode of transport for stable patients, however upskilling is to ensure timely recognition of patients requiring resuscitation or advanced care, and to determine when ambulance transport is necessary. In this study, most patients (267, 97.8%) reported access via tar roads, while a small proportion had access only through motorable mud roads (5, 1.8%) or no accessible road at all (1, 0.4%).

### 4.5. Complications and outcomes

While in-hospital deaths are typically well-documented, there is a notable lack of community-based follow-up, leading to significant gaps in post-discharge mortality surveillance—an issue particularly prevalent in low- and middle-income countries (LMICs). As a result, many deaths and complications that occur after discharge often go unreported. Of the 11 (4%)

deaths observed in this study, 9 were attributed to snakebite-related complications—2 occurred in-hospital while 4 patients had organ dysfunction at the time of discharge, and 3 left against medical advice due to a grave prognosis. The remaining 2 deaths were attributed to other causes, as those individuals had been discharged with full recovery. Regarding long-term outcomes, 89% of participants reported no current disabilities, 4% had died, and 7% were living with disabilities. These included amputation (1.5%), non-healing ulcers (1.5%), renal failure (0.4%, none requiring dialysis), and intermittent pain (3.7%) as illustrated in Fig 4. Follow-up of snakebite survivors is crucial, particularly for those with long-term complications, as they often face both physical limitations and psychological distress. Such disabilities can significantly reduce an individual's quality of life and economic productivity, underscoring the urgent need for integrated rehabilitation and support systems in snakebite-endemic regions [31].

### 4.6. Conservation practices and evidence collection

With regards to actions taken concerning the snake as shown in Fig 5, 50% of patients reported killing the snake This practice stems from outdated teachings in snakebite first response and therefore requires deliberate unlearning through updated education and community engagement on the principles of the One Health approach, which recognizes the interconnectedness of human, animal, and environmental health. The majority reported that the snake had left the scene—this group likely included both those who allowed the snake to escape and those unable to capture it. Snakes help in maintain balance in ecosystem by controlling rodent population which are a major agricultural pests and vectors of diseases. While gathering evidence can aid in species identification, current guidelines recommend capturing a clear photograph of the snake from a safe distance, rather than attempting to kill or handle it, ensuring both responder safety and the preservation of biodiversity.

### 4.7. Study limitations

1. The accuracy of responses is subject to variability in participant recall, which may affect the reliability of reported actions and challenges - particularly in retrospective accounts of pre-hospital care. Some discrepancies may stem from recall bias or intentional/unintentional misreporting. For example, when respondents reported unavailability of ambulance services, it was not always possible to distinguish whether this reflected a true lack of access or a lack of awareness about available services. However, given that snakebite is an acute and intense event, victims and first responders generally recall circumstances vividly. Furthermore, potential recall bias was mitigated by cross-examining information with available hospital medical records, VENOMS Registry entries, and in some cases pre-hospital transfer records thereby strengthening the reliability of the data.

2. The study relied on telephonic interviews, which inherently limited participant reach. Despite a larger pool of 578 potential patients, only 273 patients could be evaluated. This reduction was primarily due to unavailability of valid contact information, change of phone numbers, unanswered calls, restricted incoming calls and lack of consent.

3. Due to the remote nature of the study areas, detailed topographic mapping and systematic assessment of road accessibility to healthcare facilities could not be undertaken, which limits our ability to quantify the geographical barriers to care.

4. A substantial portion of the study period (2020–2023) coincided with the COVID-19 pandemic, during which healthcare access and health-seeking behaviours may have been disrupted. While 477 of the 578 total cases in the study occurred during this period, the study did not specifically evaluate the pandemic's impact on participant behaviour, which may have influenced patterns of pre-hospital are and facility utilization.

5. The findings of this study have limited generalizability, as healthcare infrastructure and access to emergency services vary widely across different geographic regions.

These aspects highlight the need for more robust, mixed-methods research to better understand the healthcare access landscape in snakebite-endemic regions.

### 4.8. Conclusion

This study underscores that early hospitalization is a critical determinant of improved snakebite outcomes. Key findings include the association of traditional healing practices with increased mortality; better recovery among patients who visited one or two healthcare centres compared to multiple transfers and the clear link between shorter time to first hospitalization and favourable outcomes. These results emphasize the importance of timely transfer to definitive care and minimizing delays from traditional practices or interfacility transfers. While the study did not comprehensively evaluate emergency medical services utilization, our findings suggest that strengthening ambulance availability and strengthening referral pathways may improve survival and improved outcomes.

To improve outcomes in snakebite management, several key recommendations emerge from this study.

1. Develop context-specific guidelines for pre-hospital care and emergency medical services (EMS), especially tailored to rural and remote settings. This includes standardized transfer forms, clear referral protocols, and telemedicine support for real-time decision-making, remote triage, and early initiation of appropriate management

2. Identification of high-risk areas for snakebites and the positioning of ambulances accordingly.

3. Improve accessibility by expanding ambulance types to include motorcycle- and auto-based emergency vehicles in hard-to-reach regions

4. Train informal or newly legalized transport providers to recognize severe cases early and facilitate timely referral to appropriate healthcare facilities.

### Supporting information

**S1 Text. Supporting Documents for the study.** Annexure 1: IEC approval of the study. Annexure 2: Telephonic Survey Tool. Annexure 3: Health Education Content. Annexure 4: Demographics of Non-responders. Annexure 5: Definitions of Healthcare facility and Disabilities.
(DOCX)

**S1 Data. Raw dataset used for the analysis.** This file includes the unprocessed data submitted as part of the study's submission data.
(XLSX)

### Acknowledgments

We would like to thank the Department of Emergency Medicine, Centre for Wilderness Medicine, KMC Manipal and Department of Emergency Medical Technology, Manipal College of Health Professions, MAHE, Manipal, Centre for One Health - National Centre for Disease Control, and my colleagues for their support.

### Author contributions

**Conceptualization:** Usha Wagle, Vrinda Lath, Freston Marc Sirur.

**Data curation:** Usha Wagle, Vrinda Lath, Freston Marc Sirur.

**Formal analysis:** Usha Wagle, Vrinda Lath, Vennila Jaganathan.

**Investigation:** Usha Wagle, Freston Marc Sirur.

**Methodology:** Usha Wagle, Vrinda Lath, Vennila Jaganathan, Freston Marc Sirur.

**Project administration:** Freston Marc Sirur.

**Resources:** Freston Marc Sirur.

**Software:** Usha Wagle.

**Supervision:** Vrinda Lath, Vennila Jaganathan, Freston Marc Sirur.

**Validation:** Vrinda Lath, Freston Marc Sirur.

**Visualization:** Usha Wagle.

**Writing – original draft:** Usha Wagle.

**Writing – review & editing:** Vrinda Lath, Freston Marc Sirur.

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
