## [Decision Letter · Decision Letter 0]

27 Aug 2025

Pre-hospital Interventions in Snakebite: A Telephonic Survey and follow up Investigating Snakebite envenoming from a Tertiary Care Centre in Coastal Karnataka.

Dear Dr. Sirur,

Thank you for submitting your manuscript to PLOS Neglected Tropical Diseases. After careful consideration, we feel that it has merit but does not fully meet PLOS Neglected Tropical Diseases's publication criteria as it currently stands. Therefore, we invite you to submit a revised version of the manuscript that addresses the points raised during the review process.

Please submit your revised manuscript within 60 days Oct 25 2025 11:59PM. If you will need more time than this to complete your revisions, please reply to this message or contact the journal office at plosntds@plos.org. Please include the following items when submitting your revised manuscript:

We look forward to receiving your revised manuscript.

Kind regards,

Kalana Prasad Maduwage, MBBS, MPhil, PhD, FRSPH (UK), FHEA, FRCP (Edin)

Academic Editor

José María Gutiérrez

Section Editor

Shaden Kamhawi

co-Editor-in-Chief

Paul Brindley

co-Editor-in-Chief

**Journal Requirements:**

3) We have noticed that you have a list of Supporting Information legends in your manuscript (SI,Annexure-2 and SI,Annexure-3). However, there are no corresponding files uploaded to the submission. Please upload them as separate files with the item type 'Supporting Information'.

Potential Copyright Issues:

i) Please confirm (a) that you are the photographer of 6, 7, and 8,  or (b) provide written permission from the photographer to publish the photo(s) under our CC BY 4.0 license.

ii) Figure 9 contains a logo or branding. We are not permitted to publish this under our CC-BY 4.0 license, even with permission. We ask that you please remove or replace it.

iii) Figure 9 (nail and flies image). Please confirm whether you drew the images / clip-art within the figure panels by hand. If you did not draw the images, please provide (a) a link to the source of the images or icons and their license / terms of use; or (b) written permission from the copyright holder to publish the images or icons under our CC BY 4.0 license. Alternatively, you may replace the images with open source alternatives. See these open source resources you may use to replace images / clip-art:

iv) Figure 2. Please (a) provide a direct link to the base layer of the map (i.e., the country or region border shape) and ensure this is also included in the figure legend; and (b) provide a link to the terms of use / license information for the base layer image or shapefile. We cannot publish proprietary or copyrighted maps (e.g. Google Maps, Mapquest) and the terms of use for your map base layer must be compatible with our CC BY 4.0 license.

5) We note that your Data Availability Statement is currently as follows: "All relevant data are within the manuscript and its Supporting Information files". Please confirm at this time whether or not your submission contains all raw data required to replicate the results of your study. Authors must share the “minimal data set” for their submission. PLOS defines the minimal data set to consist of the data required to replicate all study findings reported in the article, as well as related metadata and methods (https://journals.plos.org/plosone/s/data-availability#loc-minimal-data-set-definition).

6) Please provide a completed 'Competing Interests' statement, including any COIs declared by your co-authors. If you have no competing interests to declare, please state "The authors have declared that no competing interests exist". Otherwise please declare all competing interests beginning with the statement "I have read the journal's policy and the authors of this manuscript have the following competing interests:"

**Reviewers' Comments:**

Reviewer's Responses to Questions

**Key Review Criteria Required for Acceptance?**

**Methods**

-Are the objectives of the study clearly articulated with a clear testable hypothesis stated?

-Is the study design appropriate to address the stated objectives?

-Is the population clearly described and appropriate for the hypothesis being tested?

-Is the sample size sufficient to ensure adequate power to address the hypothesis being tested?

-Were correct statistical analysis used to support conclusions?

-Are there concerns about ethical or regulatory requirements being met?

Reviewer #1: Line 186-88: significant association of Tourniquet use indicating better clinical outcomes (Fisher’s Exact Test,p = 0.002) is contradictory with the set up concept of harmful intervention that WHO guidelines highilit. If patients using tourniquet increased full recovery (74.3% vs. 54.9%) with lower mortality (0.5% vs. 4.9%) compared to those who did not, this should be accordingly discussed with additional evidence. Otherwise, this will be questionable.

In my understanding, authors retrospectively phone contacted 80 snakebite patients among 273 eligible respondents. But, in Table 3, authors measured association between pre-hospital interventions adopted by patients and outcomes of treatment at hospital where they received directory of patients' contact details for phone survey. The measure of association including non-patient respondents (eyewitness) would not give detailed information on prehospital care to outcomes to reflect the impact of interventions on outcome. So, author needs to revise the entire data set to re-analyze only real-patients' data (parents in case of deceased cases and children) for impact interventions.

Similarly, the table 4 is not factual. All eye-witnesses might not accompany the patients. So, the table made based on hearsay information cannot be reliable for scientific interpretation.

Line 142-49: This section needs re-writing. Authors mentioned "no more than 15 minutes" required to get answered for set up questions (Table 1). Below, they highlighted that "Any questions participants had regarding snakebites were thoroughly addressed during the telephonic interview. Their social outreach initiative approach has weakened this section.

Reviewer #2: 1. The data on prehospital interventions should not be presented as prospectively collected, as this may be misleading.

2. Please clearly state the inclusion and exclusion criteria used for participant enrollment.

3. It would strengthen the methodology section to include a flow diagram illustrating the patient recruitment process, starting with the total number of patients registered in the VENOMS registry during the study period, followed by numbers excluded and included, the number of participants who took part in the survey, and the number who did not, along with reasons for exclusion or not participating at each step.

4. Please specify the timing of participant interviews following hospital discharge. Were the interviews conducted at a standardized time point? The timing could significantly affect reported clinical outcomes and should be clarified.

5. Please indicate who conducted the participant interviews. This information is important for assessing potential interviewer bias and overall study reliability.

Reviewer #3: Please, see summary.

**Results**

-Does the analysis presented match the analysis plan?

-Are the results clearly and completely presented?

-Are the figures (Tables, Images) of sufficient quality for clarity?

Reviewer #1: -Does the analysis presented match the analysis plan?

# needs statistical consultation first.

-Are the results clearly and completely presented?

## partly yes.

-Are the figures (Tables, Images) of sufficient quality for clarity?

## Figures needs improvement with legible texts.

Title: please, remove full stop from the title.

Abstract:

Please, replace ";" with "," while listing objectives. The objective "describe the nature of medical and non-medical pre-hospital 26 interventions;" should be further simplified.

Please, take care of capitalization throughout the manuscript e.g., A Prospective survey in abstract would be "A prospective survey ...". ..... patients, Tourniquet. See Title of this manuscript....

Similarly, assure a single space between each words, e.g., "3.38.0)and": It would be "3.38.0) and ..". response(70%) ....

Introduction

Authors need to insert citation following up the author guidelines for this journal.

Line 72-74: This "Globally, snakebite envenoming occurs in an estimated 1.8 to 2.7 million people annually, resulting in 81,000 to 138,000 deaths and approximately 400,000 cases of permanent disability." needs citation

Reviewer #2: 1. Providing basic demographic information of the study participants would be valuable for readers to better understand the characteristics of the study population.

2. Table 2: It would be more informative to present the first responses in ascending order, starting with the most common. This will help readers quickly identify the most frequently reported responses.

3. Please clarify which indicators were used to assess hemodynamic status in the study population. This is important for interpreting the clinical relevance of the findings.

4. Table 3: Consider presenting the total numbers of patients who died, fully recovered, were discharged with disability, or were discharged against medical advice in the relevant cell in the title row. This would improve the clarity and comparability of the outcome data.

5. Please avoid using non-standard abbreviations such as ED, EMT, ASHA, PHC, ALS/BLS ambulance, etc., or provide a clear legend/footnote defining them. This will ensure a better understanding, especially for international readers.

6. Tables 4 and 5: Kindly clarify the meaning of “Nil.” Explicit labeling will reduce ambiguity.

7. Figure 2: This figure does not appear to contribute significantly to the understanding of the results. The authors may wish to reconsider its inclusion and evaluate whether it adds meaningful value to the manuscript.

8. Figure 3: This again raises a concern about the timing of post-discharge interviews. What is meant by “current” in this context? Please specify the time elapsed since discharge when participants were assessed, as this can influence their reported knowledge, attitude, and outcomes.

9. Please include the guided questionnaire used for assessing current knowledge and attitudes as a supplementary document. This will enhance the transparency and reproducibility of the study.

Reviewer #3: Please, see summary.

**Conclusions**

-Are the conclusions supported by the data presented?

-Are the limitations of analysis clearly described?

-Do the authors discuss how these data can be helpful to advance our understanding of the topic under study?

-Is public health relevance addressed?

Reviewer #1: Conclusions are not fully supported by the data presented.

They described limitation in details but some limitations are so serious that introduced intolerable biases.

To discuss how these data can be helpful, Please, merge the specific recommendations based only on your study findings and discussion in the conclusion section.

Reviewer #2: The conclusions are supported by the data presented. However, it would better if the discussion could address the following facts.

1. The relevance of snake repellents in relation to health-seeking behavior is not clearly established in the current manuscript. Please elaborate on how the use or perception of repellents influenced participants’ responses or their decisions to seek health care. If the connection is indirect, consider clarifying or reevaluating its inclusion in the discussion.

2. The survey spans from 2019 to 2024, a period that includes the COVID-19 pandemic (2020–2023). The pandemic significantly disrupted daily lives (occupation, finances, travel and transportation, etc.), healthcare access, healthcare availability and health-seeking behaviors of people in most countries. A recent study from Sri Lanka (“The Impact of the COVID-19 Pandemic and Post-Pandemic Economic Crisis on Snakebite Patterns in Rural Sri Lanka,” Silva et al., 2025) highlights these effects specifically in relation to snakebite victims. It would strengthen your discussion to acknowledge and reflect on how the pandemic may have influenced your study findings, either in a dedicated paragraph in the discussion or as part of the limitations section.

Reviewer #3: Please, see summary.

**Editorial and Data Presentation Modifications?**

Reviewer #1: Major improvements are needed

305-09: This section needs improvement in support of some educational interventions (published in journals) which increased the prehospital care seeking, such as:

Vaiyapuri S, Kadam P, Chandrasekharuni G, et al. 2023. Multifaceted community health education programs as powerful tools to mitigate snakebite-induced deaths, disabilities, and socioeconomic burden. Toxicon: X, 17: 100147.

Pandey DP, Thapa CL and Hamal PK 2010. Impact of first aid training in management of snakebite victims in Madi Valley. Journal of Nepal Health Research Council, 8: 5–9.

Then, the first two paragraphs should be merged into a single paragraph.

Line 313-39: This section needs rewriting extensively without repeating results. Despite increased reliance on inappropriate first aid practices such as the use of tourniquet and the discussion of association with outcome is not matching. Not only the retrospective nature of study, but also including the past eyewitnesses (with potential memory biases) also misled the findings. So, these findings and association cannot convince the evidence-based readers.

Line 434-36: This section needs to be further highlighted in connection with formal education of first aid in schools and universities in the study regions and in India. In the formal education of first aid, there would be to orient snakebite patients to the government or private facilities where antivenoms are supplied. This would reflect a lack of both formal educaiton and public unawareness.

Reviewer #2: 1. It would be better to have an uniformity in table designs as well as in graph designs.

Reviewer #3: NA.

**Summary and General Comments**

Reviewer #1: Authors attempted to characterize Pre-hospital Interventions in Snakebite without proper research design and respondents. They need to revise this manuscript after the consultations with experts in the use of dataset to test for hypothesis from this study. There are several errors in English language use throughout the manuscript. Authors need to find and improve these errors including the structure of English usage.

Methods section is inadequate to claim the associations highlighted in the results and discussion section. Authors need statistical consultations before revising methods, results, and discussion section, and resubmitting it.

Reviewer #2: 1. Please indicate the photo courtesy.

Reviewer #3: This is a necessary study on an important and overlooked aspect of snakebite incidents, specifically the pre-medical care.

Some points need refinement by the authors:

1. The statistical analysis must be better described. Were the data analyzed for normality (which methodology)? When the authors describe results presented as mean, do they mean and SD or SE? Which tests were used to compare quantitative data?

2. Do the authors have data on the age and sex of the patients?

3. Do the authors have data on the use of AV?

4. Please compare the responders with the non-responders regarding available characteristics. Are the groups different?

5. Please provide the definitions used for classifying the order and level of healthcare facility visited, as well as the definitions used for the disabilities described.

6. Please, clarify if the Institutional Ethical Committee approved the protocol.

7. Please rewrite the discussion, avoiding describing the results again and focusing on comparing the current data with the available evidence and on the implications of the current findings.

PLOS authors have the option to publish the peer review history of their article (what does this mean? ). If published, this will include your full peer review and any attached files.

**Do you want your identity to be public for this peer review?** For information about this choice, including consent withdrawal, please see our Privacy Policy .

Reviewer #1: **Yes: ** Deb Prasad Pandey

Reviewer #2: No

Reviewer #3: No

**Figure resubmission:**
---

## [Decision Letter · Decision Letter 1]

13 Nov 2025

* A rebuttal letter that responds to each point raised by the editor and reviewer(s). You should upload this letter as a separate file labeled 'Response to Reviewers '. This file does not need to include responses to any formatting updates and technical items listed in the 'Journal Requirements' section below.

* A marked-up copy of your manuscript that highlights changes made to the original version. You should upload this as a separate file labeled 'Revised Manuscript with Track Changes '.

* An unmarked version of your revised paper without tracked changes. You should upload this as a separate file labeled 'Manuscript '.

We look forward to receiving your revised manuscript.

Kind regards,

Kalana Maduwage

Academic Editor

José María Gutiérrez

Section Editor

Shaden Kamhawi

co-Editor-in-Chief

Paul Brindley

co-Editor-in-Chief

**Journal Requirements:**

1) We note that your Data Availability Statement is currently as follows: "All relevant data are within the manuscript and its Supporting Information files, additional data sets will be provided on request prior to final publication. On acceptance entire data related to the study will be made available on Open Science Framework". Please confirm at this time whether or not your submission contains all raw data required to replicate the results of your study. Authors must share the “minimal data set” for their submission. PLOS defines the minimal data set to consist of the data required to replicate all study findings reported in the article, as well as related metadata and methods (https://journals.plos.org/plosone/s/data-availability#loc-minimal-data-set-definition).

**Reviewers' comments:**

Reviewer's Responses to Questions

**Key Review Criteria Required for Acceptance?**

**Methods**

-Are the objectives of the study clearly articulated with a clear testable hypothesis stated?

-Is the study design appropriate to address the stated objectives?

-Is the population clearly described and appropriate for the hypothesis being tested?

-Is the sample size sufficient to ensure adequate power to address the hypothesis being tested?

-Were correct statistical analysis used to support conclusions?

-Are there concerns about ethical or regulatory requirements being met?

Reviewer #1: Methods

-Are the objectives of the study clearly articulated with a clear testable hypothesis stated?

# yes

-Is the study design appropriate to address the stated objectives?

# No

-Is the population clearly described and appropriate for the hypothesis being tested?

#. The population described in this manuscript is not appropriate for hypothesis being tested.

-Is the sample size sufficient to ensure adequate power to address the hypothesis being tested?

# Not sure

-Were correct statistical analysis used to support conclusions?

# No

-Are there concerns about ethical or regulatory requirements being met?

# Not at all.

Reviewer #2: 1. Inclusion and Exclusion Criteria:

The terminology used to describe the inclusion and exclusion criteria should be made more precise. Based on the information provided, all patients registered in the VENOMS registry were RECRUITED for the study. Among these, only confirmed snakebite patients were included (INCLUSION criteria). Patients who were bitten by other animals or who did not provide consent to participate were excluded (EXCLUSION criteria). It is not necessary to list all the specific animals (non-snakebites) in EXCLUSION criteria list. Please revise the text to ensure consistency and accuracy in the use of these terms.

2. Flow Chart of Patient Recruitment:

The current flow chart depicting patient recruitment is somewhat confusing and should be reorganized for better clarity.

Suggested approach:

a. Start with the total number of patients registered during the study period in the VENOMS registry.

b. Exclude those bitten by animals other than snakes.

c. From the remaining group (snakebite patients), exclude those who could not be contacted, followed by those who declined consent.

d. The final group—patients who were contactable and gave consent—represents those included in the study.

It is not necessary to list all the specific reasons for being uncontactable. Instead, use a broader heading and present the corresponding numbers for clarity.

3. In the methodology section, please mention the information obtained by the VENOMS registry. It would prevent the confusion of OUTCOME. (Discharge outcomes and long-term outcomes). Accordingly, change the title and the description of Table 4.

Reviewer #3: See summary.

**Results**

-Does the analysis presented match the analysis plan?

-Are the results clearly and completely presented?

-Are the figures (Tables, Images) of sufficient quality for clarity?

Reviewer #1: -Does the analysis presented match the analysis plan?

# Partly, yes.

-Are the results clearly and completely presented?

## yes

-Are the figures (Tables, Images) of sufficient quality for clarity?

## yes

Reviewer #2: (No Response)

Reviewer #3: See summary.

**Conclusions**

-Are the conclusions supported by the data presented?

-Are the limitations of analysis clearly described?

-Do the authors discuss how these data can be helpful to advance our understanding of the topic under study?

-Is public health relevance addressed?

Reviewer #1: -Are the conclusions supported by the data presented?

## needs revision still

-Are the limitations of analysis clearly described?

## yes

-Do the authors discuss how these data can be helpful to advance our understanding of the topic under study?

# Yes

-Is public health relevance addressed?

#Yes

Reviewer #2: (No Response)

Reviewer #3: See summary.

**Editorial and Data Presentation Modifications?**

Reviewer #1: Although authors claimed that analysis presented aligns with the statistical consultation from Dr.Vennila, one of the co-authors and the institutional representative for statistics, I would like to suggest you to send it to a statistician for re-review of all analysis to ensure methodological and statistical accuracy. Now, this article is adequately improved requiring minor revisions.

Reviewer #2: (No Response)

Reviewer #3: See summary.

**Summary and General Comments**

Reviewer #1: General

Authors attempted to retrospectively characterize Pre-hospital Interventions of envenomed victims without proper statistical analysis. They still need to revise this manuscript with/without testing for hypothesis in a non-random dataset obtained from hospital records. For example: their justification for inferential statistics and its interpretation can mislead the results.

Specific

Title: please, remove full stop from the title in the system page of the journal, too.

Introduction

Authors still need to insert citation following up the author guidelines for this journal.

The mentioned reference number 2 is not primary source to acknowledge for "Globally, snakebite envenoming occurs in an estimated 1.8 to 2.7 million people annually, resulting in 81,000 to 138,000 deaths and approximately 400,000 cases of permanent disability.". So, please, acknowledge the primary sources for the data and text citation throughout this manuscript. Please, check whether you have acknowledged duly to all owners of the article that you have cited in this manuscript.

Methods

Method is improved well except the use of inferential statistics to generalize the findings.

Please, also mention the duration (MM-YYYY to MM-YYYY) for the telephonic survey in the data collection section.

Results

Please, improve the results for tourniquet use without associating with clinical outcomes because of retrospective data, lack of due observational evidence, and non-random dataset. To avoid questionable interpretation, please, be confined with descriptive statistical analysis with improvement in discussion accordingly.

Next, " Time taken to reach the first healthcare facility:" should be clarified whether those facilities were provided with antivenom and ancillary treatment services. In most of remote healthcare facilities, only limited amount of antivenom is supplied, which results in administration of limited dose of antivenom and referral to higher center. So, in your comprehensive survey the proportion of such facilities accessed by the enrolled patients would give further clear picture of the scenario of health services and need of improvement.

Discussion

Line 392-393: Please, improve these lines with the relevant reference citation as: "Earlier guidelines did recommend the use of tourniquets as a first-aid measure for snakebites (references for those guidelines herein, please); however, with evolving evidence, current protocols (references for those protocols herein, please) no longer advise their use. Please, improve the discussion about continuity of widely disregarded tourniquet use. This continuity might also be dependent on the erroneous first aid learning in schools and universities of Karnataka like in Nepalese contexts [see the abstract published in Toxicon: Pandey DP, Khanal BP, Sapkota H. 2024. Persistent teaching of harmful interventions for pre-hospital care of snakebites in Nepal. Toxicon 248: 108004 (108038).]. The continued use of tourniquets can be discouraged more effectively through improved formal education in this state of India.

The association of tourniquet use with improved outcomes in a telephonic survey of envenomed patients admitted in Tertiary Care Center in coastal Karnataka of India could be the resultant of unavoidable limitations in retrospective telephonic data and none of the authors observed them prospectively for the detailed information on tourniquet tightness or time of its application or its effectiveness to occlude blood flow. That is why, this is a serious concern to generalized this finding based on weak study methods. It is because the non-effective (loose) tourniquets do not restrict venom movement. The loose ligature is medically ineffective but it is less harmful. Therefore, inferential statistical interpretation of these weaker data should not be included herein. Further, systematic review performed in 2016 has highlighted no benefit from tourniquet use in snakebite as a first aid, rather higher risk of local envenoming are reported (please, see the 12th reference). So, it is better to present the retrospective data without performing the inferential analysis.

Authors' claim of more than 50% of tourniquet users (Table 5) reaching healthcare within 30 minutes does not suggest improved outcomes. Even patients admitted early in advance discharged with no envenoming and re-admitted after evoluaiton of envenoming after about 26 hours [see the reference: https://www.researchgate.net/publication/298213568]. So, early hospitalization alone is not an independent prognosticator in snakebite recovery. With this weak database for this particular aspect, you can suggest for further study to considered prospective study to achieve statistically acceptable inferences while performing the outcome analysis.

Conclusions

"In addition, 22.7% of participants reported ambulance unavailability highlighting the gaps in pre-hospital care access. While some received pre-hospital care, the nature and quality of these interventions was unclear, pointing to the need for standardized protocols". This is part of discussion section.

Please, shorten the recommendations and place them under the Conclusion section.

Reviewer #2: The investigators have responded to most of the comments raised in the previous review. However, several aspects still require attention to enhance the clarity and overall quality of the manuscript which have been mentioned under the comments for methodology.

Reviewer #3: The text has improved, but it needs a in-depth grammatical revision. There are several mistakes and typos.

Please clarify in Tables 4 and 5 where the statistical differences are found.

PLOS authors have the option to publish the peer review history of their article (what does this mean? ). If published, this will include your full peer review and any attached files.

**Do you want your identity to be public for this peer review?** For information about this choice, including consent withdrawal, please see our Privacy Policy .

Reviewer #1: **Yes: ** Deb Prasad Pandey

Reviewer #2: No

Reviewer #3: No

**Figure resubmission:**
---

## [Editor Report · Decision Letter 2]

9 Dec 2025

Dear Dr Sirur,

We are pleased to inform you that your manuscript 'Pre-hospital Interventions in Snakebite: A Telephonic Survey and follow up Investigating Snakebite envenoming from a Tertiary Care Centre in Coastal Karnataka' has been provisionally accepted for publication in PLOS Neglected Tropical Diseases.

Best regards,

Kalana Maduwage, MBBS, MPhil, PhD, FHEA, FRCP (UK), FRCP (Edin)

Academic Editor

José María Gutiérrez

Section Editor

Shaden Kamhawi

co-Editor-in-Chief

Paul Brindley

co-Editor-in-Chief

---

## [Editor Report · Acceptance letter]

Dear Dr Sirur,

We are delighted to inform you that your manuscript, "Pre-hospital Interventions in Snakebite: A Telephonic Survey and follow up Investigating Snakebite envenoming from a Tertiary Care Centre in Coastal Karnataka," has been formally accepted for publication in PLOS Neglected Tropical Diseases.

Best regards,

Shaden Kamhawi

co-Editor-in-Chief

Paul Brindley

co-Editor-in-Chief
